# Predicting Rock Brittleness Using a Robust Evolutionary Programming Paradigm and Regression-Based Feature Selection Model

**Mehdi Jamei** [1], **Ahmed Salih Mohammed** [2], **Iman Ahmadianfar** [3], **Mohanad Muayad Sabri Sabri** [4], **Masoud Karbasi** [5] **and Mahdi Hasanipanah** [6,*]

1  Faculty of Engineering, Shohadaye Hoveizeh Campus of Technology, Shahid Chamran University, Dasht-e Azadegan, Susangerd 6155634899, Iran; m.jamei@shhut.ac.ir
2  Civil Engineering Department, College of Engineering, University of Sulaimani, Sulaymaniyah 46001, Iraq; ahmed.mohammed@univsul.edu.iq
3  Department of Civil Engineering, Behbahan Khatam Alanbia University of Technology, Behbahan 6361663973, Iran; im.ahmadian@gmail.com
4  Peter the Great St. Petersburg Polytechnic University, St. Petersburg 195251, Russia; mohanad.m.sabri@gmail.com
5  Water Engineering Department, Faculty of Agriculture, University of Zanjan, Zanjan 4537138791, Iran; m.karbasi@znu.ac.ir
6  Institute of Research and Development, Duy Tan University, Da Nang 550000, Vietnam
*  Correspondence: hasanipanahmahdi@duytan.edu.vn

**Abstract:** Brittleness plays an important role in assessing the stability of the surrounding rock mass in deep underground projects. To this end, the present study deals with developing a robust evolutionary programming paradigm known as linear genetic programming (LGP) for estimating the brittleness index (BI). In addition, the bootstrap aggregate (Bagged) regression tree (BRT) and two efficient lazy machine learning approaches, namely local weighted linear regression (LWLR) and KStar approach, were examined to validate the LGP model. To the best of our knowledge, this is the first attempt to estimate the BI through the LGP model. A tunneling project in Pahang state, Malaysia, was investigated, and the requirement datasets were measured to construct the proposed models. According to the results from the testing phase, the LGP model yielded the best statistical indicators (R = 0.9529, RMSE = 0.4838, and $I_A$ = 0.9744) for modeling BI, followed by LWLR (R = 0.9490, RMSE = 0.6607, and $I_A$ = 0.9400), BRT (R = 0.9433, RMSE = 0.6875, and $I_A$ = 0.9324), and KStar (R = 0.9310, RMSE = 0.7933, and $I_A$ = 0.9095), respectively. In addition, the sensitivity analysis demonstrated that the dry density factor demonstrated the most effective prediction of BI.

**Keywords:** rock brittleness; linear genetic programming; bagged regression tree; lazy machine learning method





## 1. Introduction

The brittleness of rock should be measured as the main property of rock mass in any ground excavation project. It is important to properly consider the brittleness of the rock to design structures of geotechnical engineering, particularly structures constructed on the rock mass. For example, engineers can use the information on rock brittleness to assess the wellbore performance quality and stability of a hydraulic fracturing job [1–3]. Furthermore, such information can be used to regulate the mechanical properties of shale rocks well. Meanwhile, Young's modulus and strength of these properties can be defined using certain parameters such as the volumetric fraction of strong minerals [4–6].

One of the reasons for different disasters due to rock mechanics, such as rock bursts, is brittleness [7–9]. The literature shows that brittleness can be an effective and significant factor that can predict tunnel boring machines (TBMs) and road header performance [10,11].

Moreover, this property can effectively define the excavation effectiveness of drilling as a parameter highly affecting coal mining processes [3,12]. Therefore, measuring rock brittleness is necessary for any ground excavation project [7]. Although all of the above facts had been explained, Altindag [13] argued that there was no consensus on measurement and definition of standards for this brittleness. On the other hand, Yagiz [12] argues that rock brittleness is affected by different properties of the rock. Some researchers have described the relationship between brittleness and ductility inversion or the lack of ductility [14]. Ramsey [15] defined brittleness as the lack of cohesion in rock particles. Brittleness was defined by Obert and Duvall [16] as the inclination of a material, such as rock or cast iron, to split. There are normally six characteristics of highly brittle rock: a large compressive-to-tensile strength ratio, a large interior friction angle, the production of small particles, failure under an insignificant force, high firmness, and producing completely developed characteristics after hardness lab experiments [16].

The relationship between the rock's uniaxial compressive and tensile strengths is a significant subject in rock brittleness index (BI) studies [17–19]. Nevertheless, the relationship between BI and other rock properties such as Poisson's ratio, internal friction angle, hardness, elasticity modulus, etc. is limited in the literature [20,21]. There has not enough capability to estimate BI in these models due to them using one or two dependent parameters [12,22].

Rock brittleness can be approximated using empirical formulas proposed by several studies [20,23,24]. Alternatively, multi-input and single-input predictive methods such as multiple and simple linear regression can be used to predict the BI value of rock [22,24]. However, despite a higher accuracy than the existing simple regression [17,25], they sometimes cannot accurately describe complex systems' behavior since they are not always robust enough [26]. Furthermore, rock BI cannot be predicted due to the insufficient accuracy level of these models [22]. Recently, many researchers have applied machine learning (ML) methods and metaheuristic algorithms to solve engineering and science problems [27–33].

Despite some researchers confirming that ML techniques could be used to solve problems in engineering fields, studies with a focus on the prediction of rock BI have not used different ML techniques yet. Kaunda and Asbury [34] used Poisson's ratio, velocity, and elastic modulus to apply a neural network (NN) method. Yagiz and Gokceoglu [17] formed a fuzzy system and conducted a multiple regression method to estimate rock BI by using different input parameters such as Brazilian tensile strength (BTS). Their findings demonstrated the effective application of the fuzzy system to estimate BI. Koopialipoor et al. [25] suggested some models that predict rock BI value. The proposed equations were developed by hybridizing the firefly and ANN algorithm into a single model. Another study by Khandelwal et al. [22] tested the feasibility of a genetic programming model to predict the brittleness level of intact rocks. Multiple input variables such as unit weight, BTS, and UCS were employed to estimate the rock mass BI. Jahed Armaghani et al. [3] offered different support vector machine methods for BI prediction. In their study, different kernels were used to implement SVM methods. They indicated the effectiveness of proposed SVM methods in the BI prediction field. In another study, Yagiz et al. [28] predicted BI values through a differential evolution (DE) algorithm using 48 datasets. With this aim, they employed DE to develop linear and nonlinear models. They demonstrated an acceptable application of the DE algorithm in predicting BI. Recently, comprehensive study was conducted by Sun et al. [8] to predict BI using several efficient machine learning methods such as SVM and Chi-square automatic interaction detector methods. According to their results, the proposed models could predict BI with good performance.

This study aims to assess the applicability of a novel evolutionary programming paradigm (LGP) for estimating BI to enhance the accuracy of BI simulation compared to the previous study [3]. Three advanced machine learning methods (bootstrap aggregate (Bagged) regression tree (BRT), local weighted linear regression (LWLR), and KStar models) were implemented for evaluation of the predictive performance of the LGP approach.

To the best of our knowledge, all implemented models have not yet been used in rock mechanics-based soft computing research. Here, as the novelty, best subset analysis was employed to identify the best input combination, and the results of obtained models were validated using several metrics, a graphical tool, and error analysis. In addition, an efficient sensitivity analysis was conducted to determine the most influential features in BI modeling.

## 2. Materials and Methods

### 2.1. Materials

2.1.1. Field Investigation

A tunneling project in Pahang state, Malaysia, was used to extract the data used in this study. Additional information regarding the field study can be found in Jahed Aramaghani et al. [3]. Three tunnel boring machines (TBMs) were used to excavate 35 km of the tunnel, and drilling and blasting techniques were used to excavate the rest of the tunnel [3]. Although most of the excavated rocks consisted of granite (based on the mentioned techniques), there were metamorphic and some sedimentary rocks in the geological units. The research team collected a total of 120 granite block samples from the tunnel face at different tunnel distances and several locations, and the tests were performed by transferring these block samples to the rock mechanics laboratory. Then, the procedure suggested by the ISRM [35] was applied to prepare the block rock samples for each planned test. Laboratory tests—including UCS, point load, density, the Schmidt hammer, BTS, and p-wave—were planned and conducted on the samples in the experimental program. Then, to model this study, the obtained results were considered. As mainly suggested by the literature, the BI values were calculated as BI = UCS/BTS, and then set as the output. The related inputs of the model included the p-wave velocity ($V_p$), point load strength index ($Is_{50}$), dry density (D), and Schmidt hammer rebound number ($R_n$). In Figures 1 and 2, BTS and UCS tests were conducted on the samples and their failures, respectively.

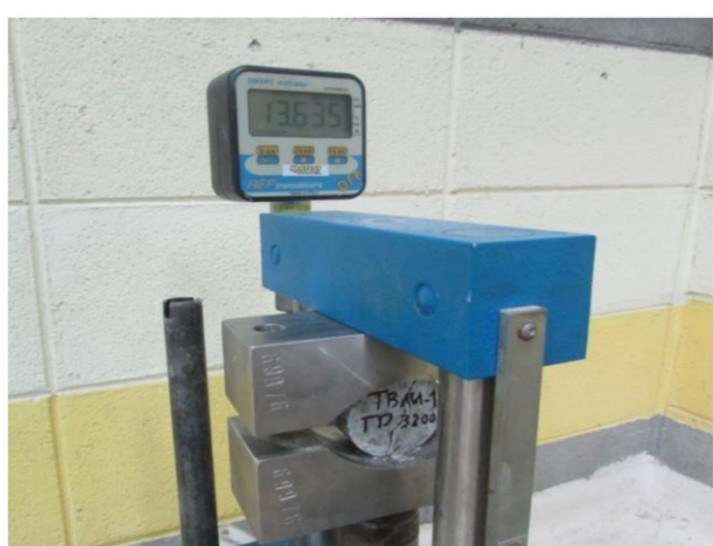

**Figure 1.** Failure of a sample under a BTS test [3].

In this study, 85 data points were collected to model the BI; 75% (64 data points) of the data was allocated for the training dataset, with the rest for the testing dataset. The descriptive statistics of all features and target variables are tabulated in Table 1. The skewness ([0.116, 0.7339]) and kurtosis ([−0.76, 0.3369]) range of variables demonstrate that both criteria fall in an acceptable range ([−2, 2]) [36,37]. Thus, it can be inferred that all datasets have a fairly normal distribution, which is a good indication for modeling rock brittleness with data-driven methods.

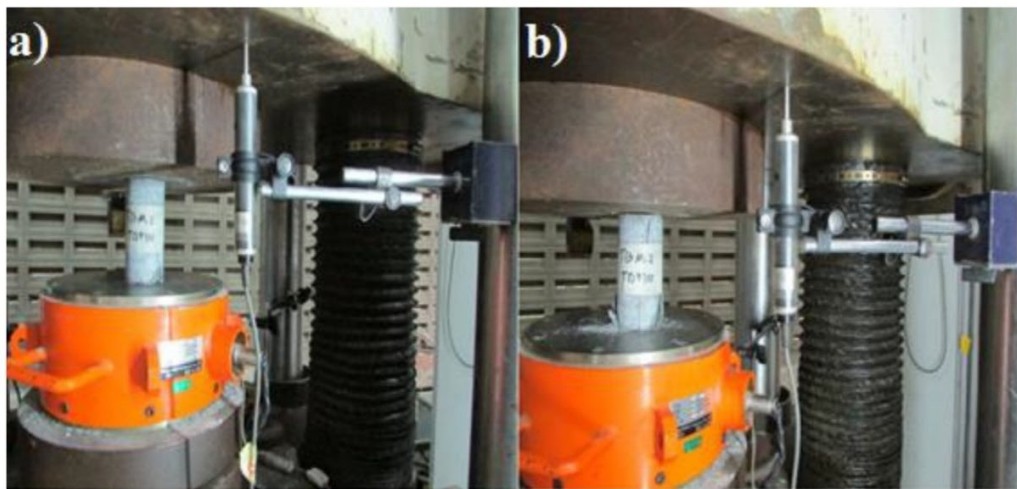

**Figure 2.** UCS test (**a**) before and (**b**) after failure [3].

**Table 1.** Descriptive statistics of all variables used in the modeling.

| Parameters | $R_n$ | $V_p$ (m/s) | Dry Density (g/cm$^3$) | $Is_{50}$ (MPa) | BI |
|---|---|---|---|---|---|
| Minimum | 20 | 2910 | 2.38 | 0.8722 | 10.12 |
| Maximum | 59 | 7943 | 2.75 | 6.59 | 16.75 |
| Mean | 37.16 | 4975 | 2.536 | 3.441 | 12.61 |
| Std. Deviation | 10.12 | 1199 | 0.079 | 1.118 | 1.554 |
| Range | 39 | 5033 | 0.37 | 5.718 | 6.626 |
| Skewness | 0.3951 | 0.2449 | 0.1161 | 0.1294 | 0.7339 |
| Kurtosis | −0.76 | −0.605 | −0.3473 | 0.3369 | 0.2216 |

The predictive and target parameters for decreasing the computational cost and complexity of prediction, normalized in range of [0, 1], are expressed in the following formula:

$$x_{nor} = \frac{x - x_{min}}{x_{max} - x_{min}} \tag{1}$$

where the $x_{nor}$ is the normalized value and $x_{max}$, $x_{min}$, and $x$ are the maximum, minimum, and original values of the modeling dataset, respectively.

### 2.1.2. Feature Selection Process

Best feature selection is one of the most crucial stages for building a predictive model based on a data-driven model; it has a key role in the accuracy and reliability of developed models. Best subsect regression analysis [38] is one of the most popular schemes for identifying the best input features based on linear regression modeling. In this approach, six metrics (mean square error (MSE), correlation coefficient (R), adjusted R$^2$, Mallows coefficient (Cp) [39], Akaike (AIC) [40], and Amemiya (PC) [41]) have been computed for choosing the best input combination [38] (see Table 2). The possible tree combination demonstrates that the last case includes all input parameters and has the highest R$^2$ (0.817) and lowest Mallows, Akaike, and Amemiya (MSE = 0.463, Cp = 5 AIC = −60.552, and PC = 0.21); as such, this case can be identified as the best combination for modeling BI. Thus, the functional relationship between the chosen features and target can be expressed as follows:

$$BI = f(R_n, V_p, D, I_{S50}) \tag{2}$$

**Table 2.** Best subset analysis for selecting the optimum input combination.

| Number of Variables | Variables | MSE | R² | Adjusted R² | Mallows' Cp | Akaike's AIC | Amemiya's PC |
|---|---|---|---|---|---|---|---|
| 2 | $V_p/D$ | 0.652 | 0.736 | 0.730 | 36.387 | −33.419 | 0.276 |
| 3 | $V_p, D, I_{S50}$ | 0.530 | 0.788 | 0.781 | 15.611 | −50.109 | 0.227 |
| 4 | $R_n, V_p, D, I_{S50}$ | 0.463 | 0.817 | 0.808 | 5.000 | −60.552 | 0.201 |

*2.2. Methods*

2.2.1. Linear Genetic Programming (LGP)

The LGP is a novel variant of the GP model proposed by Koza [42]. The LGP model is a version of the tree-based GP model with linear instruction. A comparison between the structure of the LGP and GP models is displayed in Figure 3. In the LGP, each program is described by using a parameter-length sequence of C language instructions. The instructions of LGP model include arithmetic operations (+, −, ÷, ×), conditional branches (if $x[i] \leq y[l]$), and function calls ($\exp(x)$, $x$, sin, cos, tan) [43]. Each function consists of an assignment to a parameter $x[i]$, which simplifies the utilization of multiple outputs in the LGP model. Table 3 reports the functional set and operation parameters employed in the GP. The main steps of the LGP can be described as follows:

A. **Initialization**: Creating the initial population randomly (programs), and then calculating the fitness function of each program.

B. **Main operators**:

    (1) **Tournament selection**: This operator randomly selects several individuals from the population. Two individuals with the best fitness functions are chosen from these individuals, and two others as the worst solutions [43].

    (2) **Crossover operator**: This operator is applied to combine some elements of the best solutions with each other to create two new solutions (individuals).

    (3) **Mutation operator**: Mutation is used to create two new individuals by transforming each of the best solutions.

C. **Elitist mechanism**: The worst solutions are replaced with transformed solutions based on this mechanism.

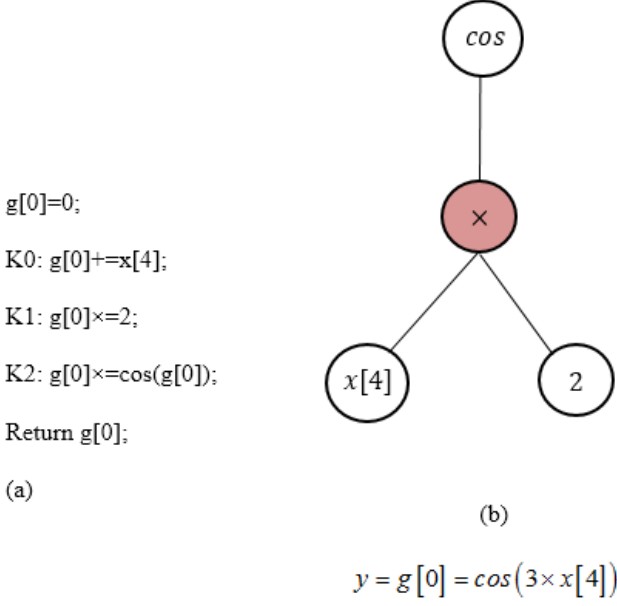

$$y = g[0] = cos(3 \times x[4])$$

**Figure 3.** Comparison between (**a**) GP and (**b**) LGP structure.

**Table 3.** The characteristics and setting parameters of proposed AI-based approaches.

| Models | Setting of Parameter | |
|---|---|---|
| LGP | Function set | +, −, ×, ÷, $\sqrt{\ }$, power, sin, cos |
| | Population size | 300 |
| | Mutation frequency % | 85 |
| | Crossover frequency % | 50 |
| | Number of replication | 10 |
| | Block mutation rate % | 20 |
| | Instruction mutation rate % | 20 |
| | Instruction data mutation rate % | 60 |
| | Homologous crossover % | 90 |
| | Program size | 64–256 |
| LWLR | ● $\mu = 4$ | |
| KStar | ● Global blend = 30 | |
| BRT | ● Function: "Bag", Learning cycles = 50, MinLeafSize = 1 | |

2.2.2. Local Weighted Linear Regression (LWLR)

The LWLR method is an advanced version of the multiple linear regression (MLR) model developed by Atkeson et al. [44]. LWLR is able to improve MLR performance significantly. To illustrate the LWLR model, consider the following model:

$$z_{mk} = \alpha_{ko} + \sum_{m=1}^{M} \alpha_{km} x_{km} + \varepsilon_k \tag{3}$$

In the above model, $z_{mk}$ is a dependent variable that can be calculated based on at least two independent variables ($x_k$). $\alpha$ is the regression coefficient calculated by the least-squares (LS) method, $M$ is the number of data, and $\varepsilon$ is the random error.

In the LWLR method, a weight function describes the relationship between input and output data. The fitness function of the LWLR model can be expressed by the following equation [44–46]:

$$F = \frac{1}{2M} \sum_{m=1}^{M} w_m (z_{om} - z_m)^2 \tag{4}$$

where $w$ is the regression weight, $z_o$ is the observed data, and $z$ is the data obtained from the model. The above equation can also be expressed in the form of the following matrix:

$$F = (X\alpha - Z)^T W (X\alpha - Z) \tag{5}$$

By solving the above equation for $\alpha$, we obtain

$$\alpha = \left( X^T W X \right)^{-1} X^T W Z \tag{6}$$

where $X$ is the matrix of input training dataset, $W$ denotes the weight matrix, and $Z$ is the vector of data obtained from the model. A kernel function can be used instead of the weight matrix in the LWLR model [47,48]. In the present study, the RBF function was used as the kernel in the LWLR model. The RBF kernel equation is defined as follows:

$$w_{ik} = \exp\left( -\mu (x_i - x_k)^2 \right) a \tag{7}$$

where $\mu$ is a positive number as a kernel variable and $(x_i - x_k)$ is the difference between point $i$ and $k$ [49]. It should be mentioned that the main setting parameter of LWLR model can be optimized by a trial and error procedure.

### 2.2.3. KStar Model

The KStar algorithm is a lazy learner method introduced by Machine [50]. This method is an instance-based (IB) algorithm with a fast learning capability. Generally, the IB requires only one instance for each group to create successful estimations. In this method, the distance between various instances is considered by the complexity of transforming an instance into another [51]. The KS employs an entropy-based distance function for the regression.

Considering a transformation and instance as $V$ and $I$, respectively, the instance maps to other instances utilizing $i : I \to I$ which belong to $V$ ($i \in V$). For mapping instances to themselves, a parameter called the distinct member ($\mu$) is used, where $\mu(\alpha) = \alpha$. This parameter is used to determine all prefix codes from $V^*$. $V^*$ comprises members which describe a one-to-one transformation to $V$. Provided that the $Pf$ is a probability function on $V^*$, the probability of all paths from $n$ to $m$ is defined as

$$P^* \left( \frac{m}{n} \right) = \sum P(v) \tag{8}$$

where $v$ is the value of set $V$. Then, the $K^*$ function can be expressed as

$$K^* \left( \frac{m}{n} \right) = -\log_2 P^* \left( \frac{m}{n} \right) \tag{9}$$

If the examples are real numbers, then it is possible to demonstrate that $P^* \left( \frac{m}{n} \right)$ is dependent solely on the absolute difference between $m$ and $n$. Therefore, it can be defined as

$$K^* \left( \frac{m}{n} \right) = K^*(i) = \frac{1}{2} \log_2 \left( 2e - e^2 \right) - \log_2(e) + i \left[ \log_2(i - e) - \log_2 \left( 1 - \sqrt{2e - e^2} \right) \right] \tag{10}$$

where $i = |m - n|$ and $e$ denotes the model parameter, whose possible values range from 0 to 1. As a result, the distance between two points is equivalent to their absolute difference. Furthermore, for real numbers, the assumption is that the real space is underlain by a discrete space with extremely short distances between discrete instances. The first thing that has to be done is to evaluate those expressions in their limit as the variable $e$ becomes closer and closer to 0. Thus, we obtain

$$P^*(i) = \sqrt{e/2} \cdot e^{-i\sqrt{2e}} \tag{11}$$

The likelihood of generating an integer with a value between $i$ and $I + i$ can be expressed as a probability density function (PDF) as follows:

$$P^*(i) = \sqrt{e/2} \cdot e^{-i\sqrt{2e}} \cdot \Delta i \tag{12}$$

To obtain the PDF over the real numbers, $x/x_0 = i\sqrt{2e}$ can be adjusted in aspects of a real value $x$.

$$P^*(x) = \frac{1}{2x_0} e^{\frac{-x}{x_0}} dx \tag{13}$$

where $x_0$, the mean predicted value for $x$ across the distribution $P$, must be suitable for practical purposes. A number between $n_o$ and $N$ is picked in the KStar method, which selects $x_0$ as the training instance with the lowest distance from $m$. It should be noted that the KS model was developed in this study by utilizing open-source WEKA software. The main parameter of the KS model is the global blend (GB: $n$), which is determined by using the trial-and-error method.

#### 2.2.4. Bootstrap Aggregate (Bagged) Regression Tree (BRT)

Bagging (bootstrap aggregating) is one of the learning methods of the ensemble learning model [52]. In this method, the training data series is divided into *N* new training data series by the bootstrap sampling method, and a weak learner is used to train *N* datasets. In the bootstrap sampling method, random sampling is performed by replacement, which means that some of the training series data may be repeated, and some may be omitted. In the bagging regression tree (BRT) method, each of the *N* training data series is learned by a tree regression model. The final result is obtained by averaging the output of the *N* tree models (Figure 4). In the tree regression method, the results of each tree individually have high variance and low bias. Averaging the results of *N* trees reduces the variance of the model, increases the accuracy, and prevents overfitting of the model. The performance of the BRT method depends on the correct choice of the number of trees (*N*). To determine the optimal value of *N*, out-of-bag (OOB) error estimation curves can be used. Usually, two-thirds of the data series are used in model training by bootstrapping. One-third of the remaining data that does not enter the training phase in each tree is called out-of-bag (OOB) observations. OOB observations are used to estimate the prediction error. The error value of the obtained OOB observations is a good criterion for model error validation. In the present study, the fit ensemble function in MATLAB software was used to build a bagged regression tree model.

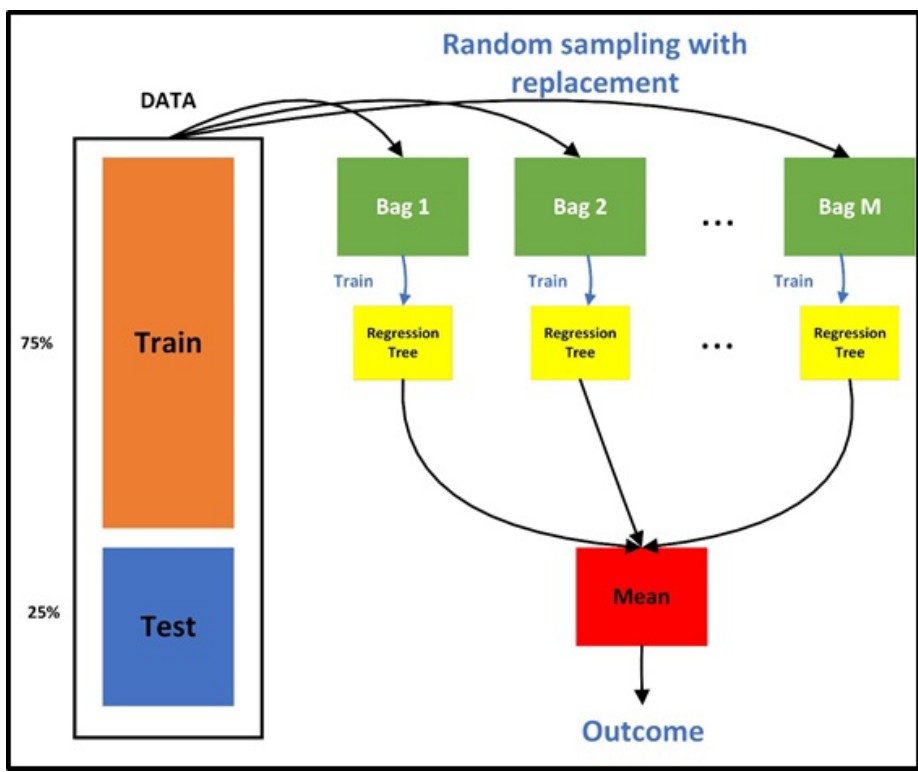

**Figure 4.** Training procedure of bagged regression tree (BRT).

#### 3. Statistical Criteria for Evaluation of Models

To check the precision of the proposed models, different statistical criteria including *R*, root mean square error (RMSE), mean absolute percentage error (MAPE), Scatter Index (SI), and Willmott's agreement Index (IA) were employed [53–66].

1.   Correlation coefficient (R) can be expressed as

$$R = \frac{\sum_{i=1}^{N}\left(BI_{p,i} - \overline{BI}_p\right).\left(BI_{o,i} - \overline{BI}_o\right)}{\sqrt{\sum_{i=1}^{N}\left(BI_{p,i} - \overline{BI}_p\right)^2 \sum_{i=1}^{N}\left(BI_{o,i} - \overline{BI}_o\right)^2}}, \ 0 < R < 1 \tag{14}$$

2. Root mean square error (RMSE) can be expressed as

$$RMSE = \left( \frac{1}{N} \sum_{i=1}^{N} \left( BI_{o,i} - BI_{p,i} \right)^2 \right)^{0.5} \tag{15}$$

3. Mean absolute percentage error is defined as

$$MAPE = \frac{100}{N} \sum_{i=1}^{N} \frac{\left| BI_{o,i} - BI_{p,i} \right|}{BI_{o,i}} \tag{16}$$

4. Scatter Index can be expressed as

$$SI = RMSE / \overline{BI}_o \tag{17}$$

5. Willmott's agreement Index [49] can be expressed as

$$I_A = \frac{\sum_{i=1}^{N} \left( BI_{o,i} - BI_{p,i} \right)^2}{\sum_{i=1}^{N} \left( \left| BI_{o,i} - \overline{BI}_o \right| + \left| BI_{o,i} - \overline{BI}_o \right| \right)^2}, \ 0 < I_A < 1 \tag{18}$$

where $BI_o$ is observed value; $BI_p$ is predicted value; $\overline{BI}_o$ and $\overline{BI}_p$ are average values of observed and predicted data, respectively; and N is the number of data.

## 4. Results and Discussion

The LGP model is provided based on free software called "Discipulus"; its setting parameters are listed in Table 3. In addition, to provide the BRT model, the "bag" method of the "fitresemble" function of the Machine Learning Toolbox of MATLAB 2019 was implemented. The setting parameters for the BRT model are tabulated in Table 3, which were optimized to avoid overfitting by using a trial-and-error procedure [67,68]. The kernel variable in the LWLR model was adopted through a trial-and-error process, leading to a value of 0.4. To provide the KStar model, the global blend—as a crucial parameter of the model—was optimized using a grid search scheme, leading to a value of 30. Figure 5 demonstrates the road map of predicting the procedure of BI parameters using provided AI models.

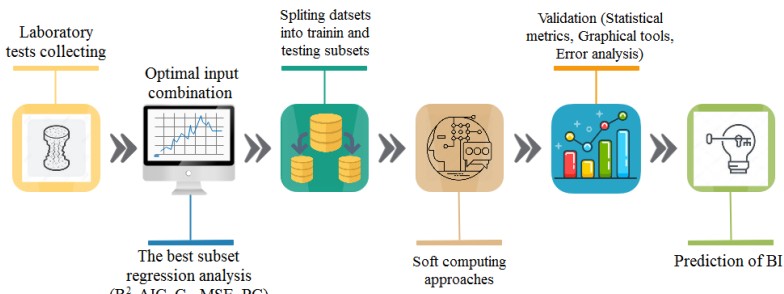

**Figure 5.** The road map of predicting BI using machine learning approaches.

This paper examines the LGP approach to predict the brittleness index (BI) based on four input variables: $R_n$, $V_p$, D, and $Is_{50}$. Also, two lazy machine learning models (namely LWLR and KStar) and a tree decision-based model (BRT) were measured to evaluate the outcome of the LGP approach. Figure 6 depicts the regression tree constructed from the BRT model, in which the terminal nodes or leafs identify the response of prediction. Table 4 presents the modeling results obtained by all models in the training and testing phases. The quantitative results in the training phase indicate that the KStar model (R = 0.9984, RMSE = 0.0865, MAPE = 0.2564, and $I_A$ = 0.9992) is superior to the BRT (R = 0.9459, RMSE = 0.5297, and MAPE = 3.1569), LWLR (R = 0.9252, RMSE = 0.5960, and MAPE = 3.4088), and LGP (R = 0.9248, RMSE = 0.5867, and MAPE = 3.6279) models. Testing

results show that the LGP approach exhibits the best efficiency for BI prediction by having the highest correlation coefficient (R = 0.9529) and lowest metrics error (RMSE = 0.4838 and MAPE = 3.2155), followed by LWLR (R = 0.9490, RMSE = 0.6607, and MAPE = 4.1549), BRT (R = 0.9433, RMSE = 0.6875, and MAPE = 4.3884), and KStar (R = 0.9310, RMSE = 0.9733, and MAPE = 5.0573), respectively. A scatter plot of each model, as a powerful graphical tool, is depicted in Figure 7 for comparison between predicted and observed values of BI. Careful examination of the scatters indicates that the LGP approach—due to the closest distribution of predicted points to the 1:1 line—demonstrates better performance than the other AI methods for whole data. The LWLR and BRT models, with acceptable accuracy and similar predictive performance, are ranked as the second and third best models, respectively. KStar, despite the remarkable performance in the training phase (R = 0.9984), is identified as the weakest method due to the highest dispersion of testing predicted points.

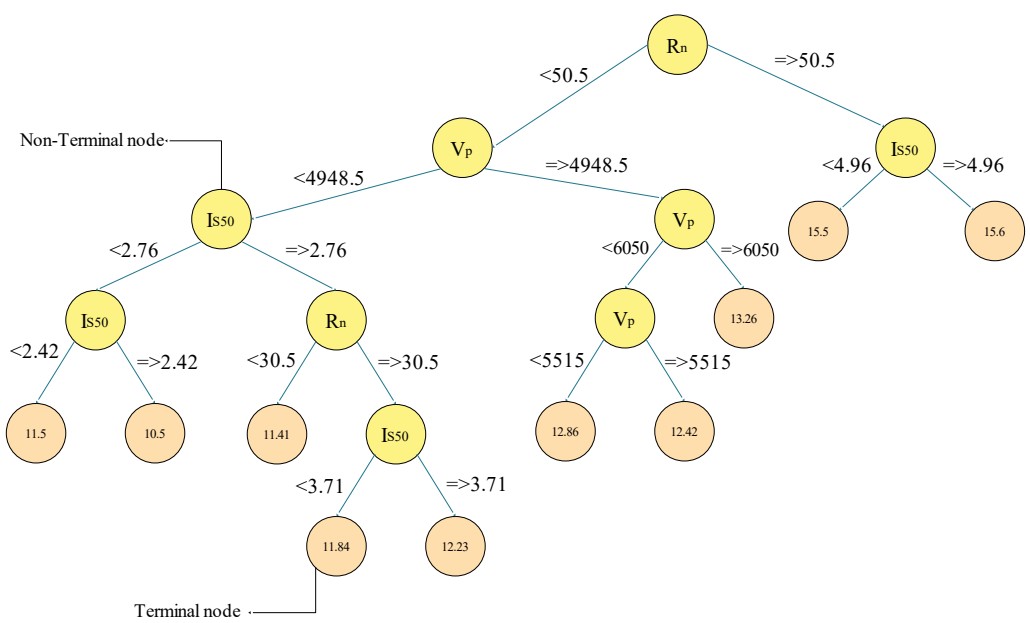

**Figure 6.** Decision trees of BRT model for prediction of BI.

**Table 4.** Quantitative evaluation of AI base approaches for predicting BI.

|          | Metrics | LGP | K-Star | BRT | LWLR |
|----------|---------|--------|--------|--------|--------|
| Training | R       | 0.9248 | 0.9984 | 0.9459 | 0.9252 |
|          | RMSE    | 0.5867 | 0.0865 | 0.5297 | 0.5960 |
|          | MAPE%   | 3.6279 | 0.2564 | 3.1569 | 3.4088 |
|          | SI      | 0.0463 | 0.0068 | 0.0418 | 0.0470 |
|          | $I_A$   | 0.9560 | 0.9992 | 0.9628 | 0.9531 |
|          | St.D    | 1.3339 | 1.5195 | 1.2640 | 1.2828 |
| Testing  | R       | 0.9529 | 0.9310 | 0.9433 | 0.9490 |
|          | RMSE    | 0.4838 | 0.7933 | 0.6875 | 0.6607 |
|          | MAPE%   | 3.2155 | 5.0573 | 4.3884 | 4.1549 |
|          | SI      | 0.0389 | 0.0638 | 0.0553 | 0.0532 |
|          | $I_A$   | 0.9744 | 0.9095 | 0.9324 | 0.9400 |
|          | St.D    | 1.5059 | 1.0861 | 1.1116 | 1.1686 |

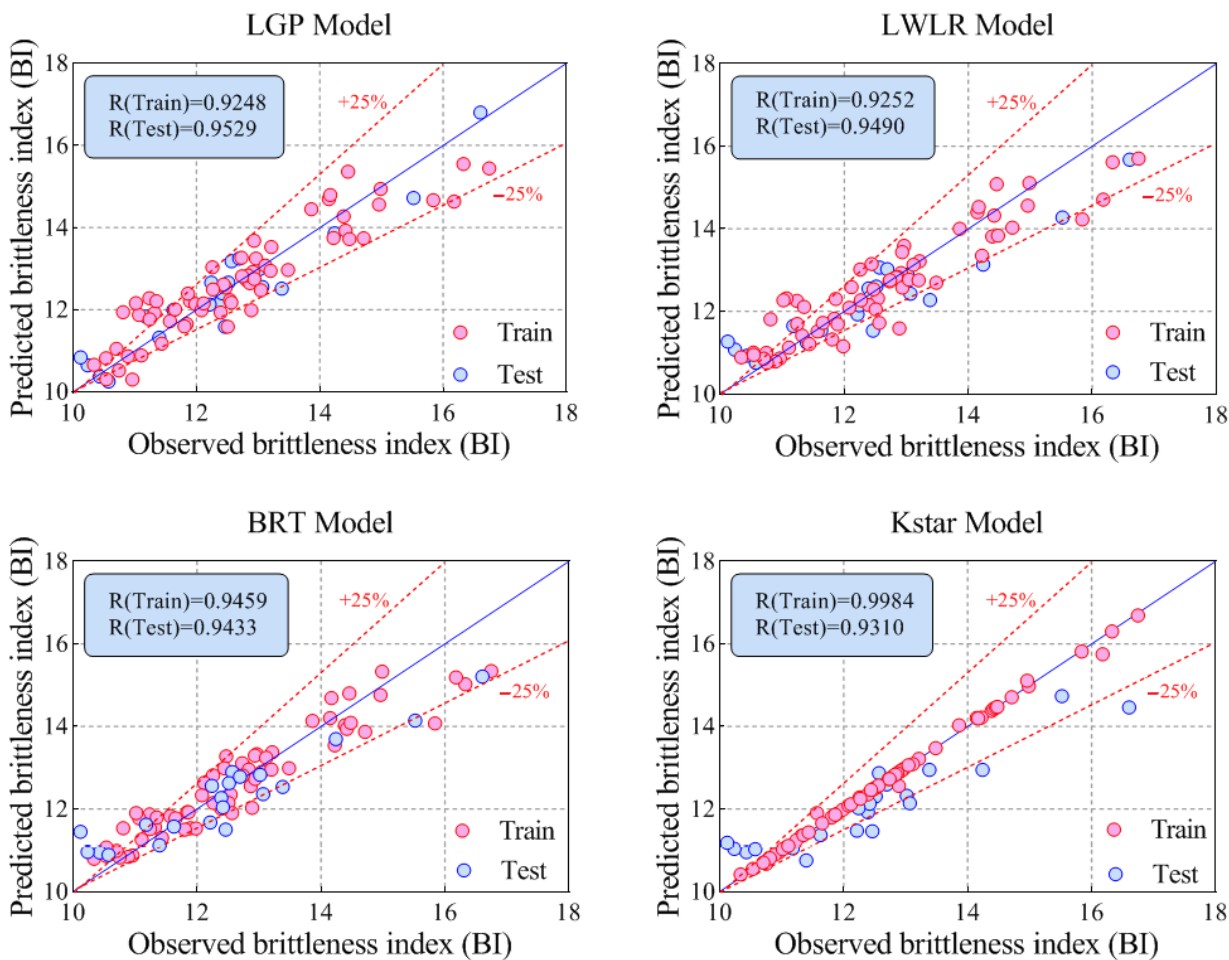

**Figure 7.** Comparison of four soft computing approaches and observed BI using scatter plot.

In the next graphical validation stage, half violin plots for all datasets are featured to show the distribution of quantitative data across several predicted values levels compared to observed ones. The underlying distribution of the models has been estimated using a smooth kernel density function by showing attractive benchmark points, namely the median and quartiles depicted in Figure 8. It is abundantly clear that KStar and BRT have closer Q25% values (11.366 and 11.562, respectively) to the observed values (11.395) compared with the LWLR and LGP approaches, whereas the LGP and LWLR Q75% values (13.251 and 13.146, respectively) exhibit better agreement with the observed values (13.21). Given the arrangement of the datasets, it is evident that the first Q25% is filled into the training data. Regarding KStar, the remarkable performance in training and disappointment in the testing phase implies that overfitting occurred in this paradigm.

The trend variation of BI plots in both training and testing modes is shown in Figure 9. The results indicate that the LGP model can properly capture the nonlinear behavior of BI in both triaging and testing datasets, and is capable of demonstrating promising predictive performance compared to other models. Complete error analysis was performed to evaluate the performance of the proposed predictive methods in BI estimation. According to Figure 10, the KStar (RDB = 5.52%) and LWLR (RDB = 21.51%) models are identified as having the best and worst predictive performance, as indicated from the lowest and highest relative deviation bands in the training stage, respectively. Furthermore, LGP with the lowest RDB (14.40%) and KStar with the highest RDB (23.41%) have yielded the most promising and weakest results in forecasting BI in testing mode, respectively.

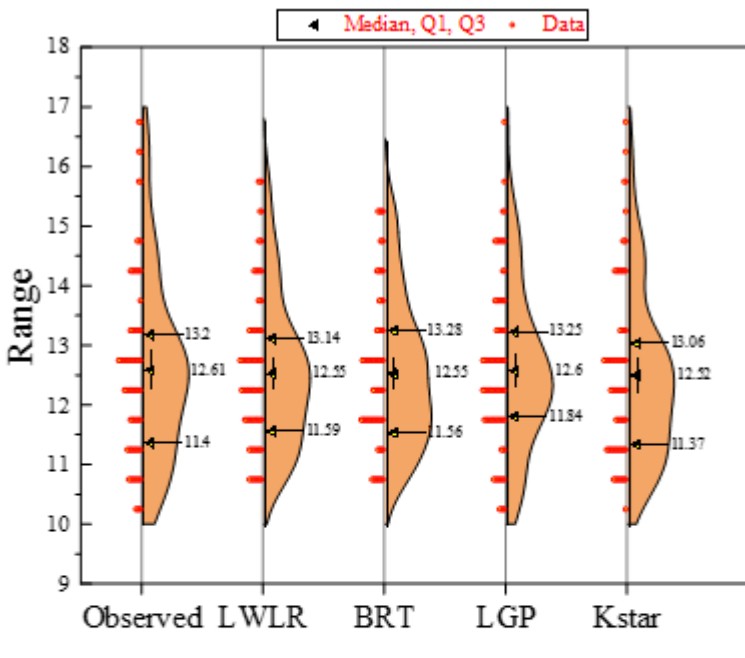

| Variable | LWLR | BRT | Kstar | LGP | Observed |
|---|---|---|---|---|---|
| $Q_{0.25}$ | 11.594 | 11.562 | 11.366 | 11.843 | 11.395 |
| Median | 12.552 | 12.553 | 12.521 | 12.607 | 12.610 |
| $Q_{0.75}$ | 13.146 | 13.287 | 13.062 | 13.252 | 13.21 |
| IQR | 1.572 | 1.725 | 1.696 | 1.409 | 1.815 |

**Figure 8.** Performance assessment of predicted and observed BI values using half violin plots.

As a final error assessment, the cumulative distribution function (CDF) of the absolute percentage of relative error (APRE) for the testing dataset was considered. Figure 11 indicates that for more than 80% of testing datasets in predicting BI values, the APRE values of LGP, BRT, LWLR, and KStar are less than 5%, 7.01%, 7.65%, and 7.80%, respectively. Eventually, it can be inferred that the LGP model, as the main novelty of this research, is superior to all proposed AI models for accurately predicting BI. The KStar approach, despite its amazing performance in the training phase, yielded the weakest results in the test phase among all models, which means that this method may not work properly for unseen data. The KStar model cannot be identified as an efficient predictive method for BI prediction due to overfitting. Thus, LGP and LWLR were identified as the best and second-best predictive models. The BRT model—ranking third, with predictive performance close to LWLR—yielded the admitted results for the prediction of BI values. It is worth noting that although KStar in this study showed unfavorable performance in testing mode, the accuracy of its results is far better than the results of previous research [3]. In the literature, some studies have predicted BI by using different machine learning methods. Yagiz et al. [69] used the genetic algorithm (GA) and particle swarm optimization (PSO) to predict BI. According to their results, the values of $R^2$ ranged between 0.851 and 0.932. In another study, Koopialipoor et al. [25] predicted BI through a combination of ANN and firefly algorithm, yielding prediction results with an $R^2$ of 0.896. In the present study, BI has been predicted with better performance ($R^2$ of 0.953) from the LGP model. This indicates the effectiveness of the model proposed in this study compared to aforementioned models

used in the literature. According to the objectives of this study, the uncertainty of the data has not been investigated. Given great importance, uncertainty of data and results of machine learning-based methods could be considered as the subject of future research. Also, the models presented in the current study generally suffer from a lack of laboratory data. Therefore, in the future, it is necessary to examine the accuracy of presented methods with a greater number of datasets.

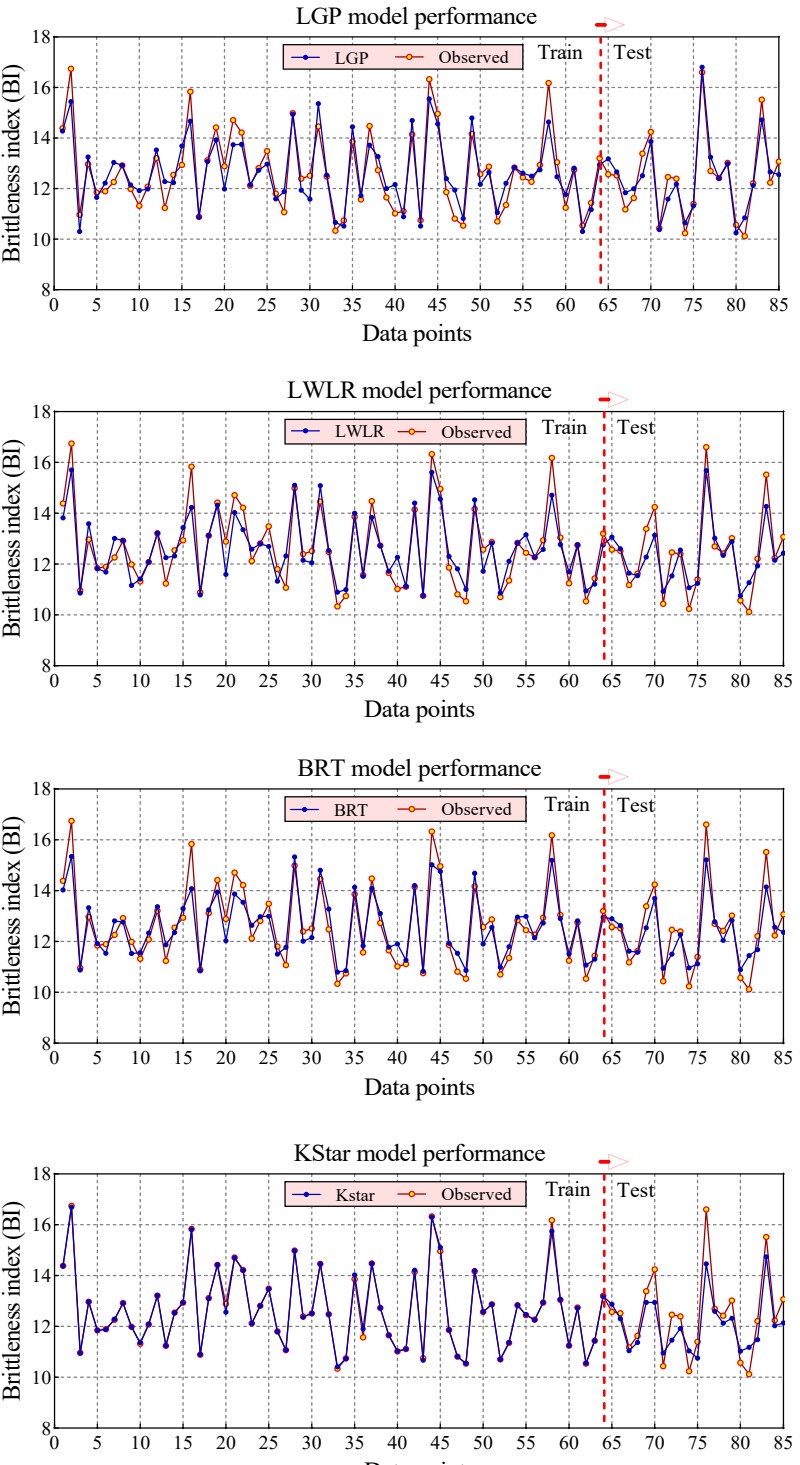

**Figure 9.** The trend physical plot for comparison between the observed and predicted BI values.

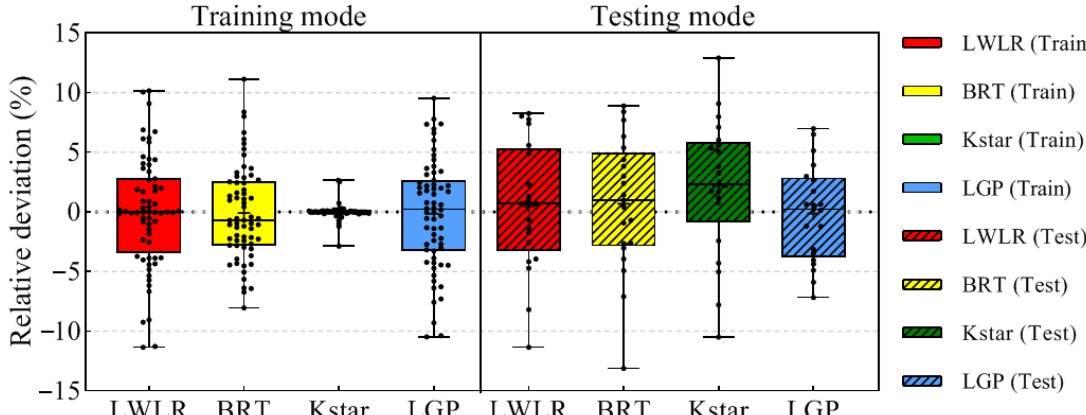

**Figure 10.** Box plots for the relative deviation (%) distribution of all predictive models in testing and training.

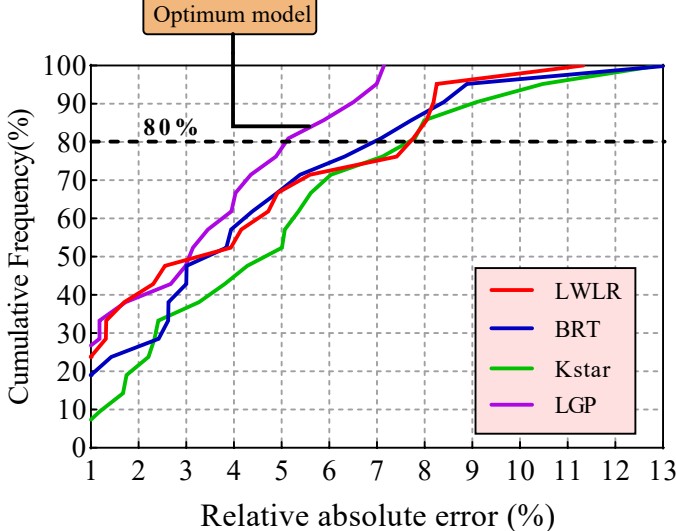

**Figure 11.** The cumulative frequency percentage versus the relative absolute error (%) for LWLR, BRT, KStar, and LGP models for the testing dataset.

## 5. Sensitivity Analysis

For more effective use of the AI methods, recognizing the effective parameters is an essential issue. One of the most widely used techniques for sensitivity analysis (SA) is consecutive elimination of the input variables and executing the AI model for all created situations. This research used the LGP model as the best model to implement the SA. Table 5 lists the SA results for five modes of combining inputs. The results demonstrate that Dry Density, with the lowest R (0.9081) and highest RMSE (0.8027) and MAPE (5.4642), is the most efficient input variable to estimate the brittleness index (BI). In addition, the $V_p$ (R = 0.9163 and RMSE = 0.7944) ranks second, followed by $Is_{50}$ (R = 0.9169 and RMSE = 0.7861) and $R_n$ (R = 0.9273 and RMSE = 0.6959). A spider plot based on the six statistical criteria for all combining inputs is displayed in Figure 12. According to this figure, the combination with eliminating the dry density variable (i.e., all-dry density), showing the lowest R and $I_A$ and highest RMSE and MAPE, has the greatest impact on the accuracy of predicting BI. It should be mentioned that some feature selection methods such as Boruta-random forest can be utilized to specify the influential parameters, which has great ability to capture the non-linear interaction between the predictors and target. This aim can be considered as an alternative of classical sensitivity analysis.

**Table 5.** The sensitivity analysis results for all possible situations.

| Metric | All-$R_n$ | All-$V_p$ | All-Dry Density | All-Is$_{50}$ | All |
|---|---|---|---|---|---|
| R | 0.9273 | 0.9163 | 0.9081 | 0.9169 | 0.9433 |
| RMSE | 0.6959 | 0.7944 | 0.8027 | 0.7861 | 0.6875 |
| MAPE | 4.4592 | 5.1695 | 5.4642 | 5.0433 | 4.3884 |
| SI | 0.0560 | 0.0639 | 0.0646 | 0.0633 | 0.0553 |
| $I_A$ | 0.9318 | 0.9018 | 0.9004 | 0.9049 | 0.9324 |
| St.D | 1.6277 | 1.6277 | 1.6277 | 1.6277 | 1.6277 |
| Rank | 4.0000 | 3.0000 | 1.0000 | 2.0000 | - |

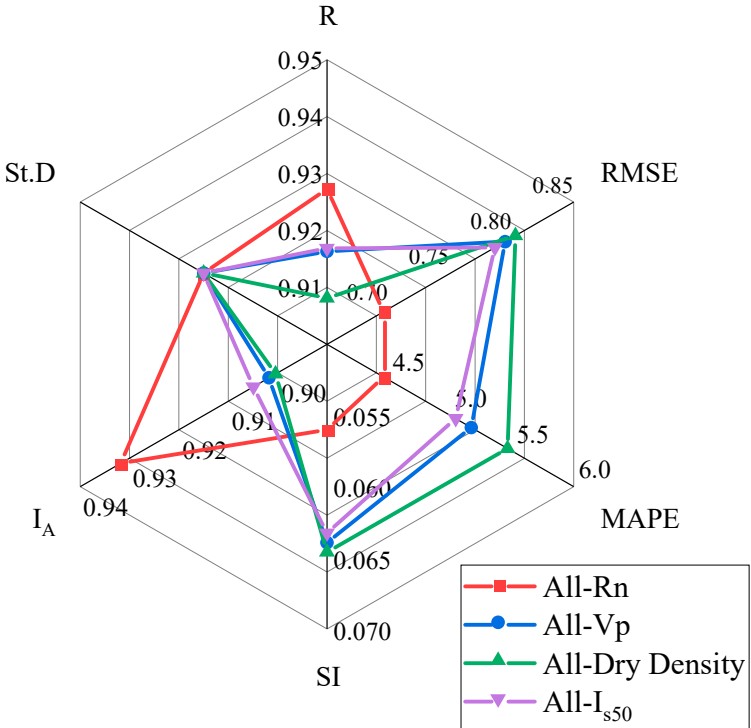

**Figure 12.** The influence input variables ranking for estimation of BI value.

## 6. Conclusions

Precise estimation of BI is necessary for any ground excavation project, and this issue requires the application of appropriate prediction models. With this in view, several advanced ML methods, including LGP, BRT, LWLR, and KStar models, were proposed to estimate BI. In this regard, a database collected from a tunneling project in Pahang state, Malaysia, was used, using four input parameters ($V_p$, Is$_{50}$, D, and $R_n$) and BI as the output parameter. In the modeling processes, 64 and 21 datasets, respectively, were used for training and testing phases. Finally, the models' accuracy was compared using several statistical criteria such as R and RSME. The findings of this study can be summarized as follows:

1.  Based on the results, all developed models' performance capacity was suitable and acceptable. Accordingly, all proposed models can be used with confidence for future research on predictions of other issues in the field of rock mechanics.
2.  Among the proposed models, the KStar (R = 0.9984 and RMSE = 0.0865) model predicted BI with the best performance in the training phase, while the best performance for the testing phase was achieved by the LGP (R = 0.9529 and RMSE = 0.4838) model. In addition, both LWLR (R = 0.9490 and RMSE = 0.6607) and BRT (R = 0.9433 and RMSE = 0.6875), ranking second and third, respectively, lead to desired results for modeling BI values.

3. The authors recommend increasing the accuracy of BI modeling as a possible future study, examining the ensemble of stacked models to integrate the advantages of standalone data-driven models.

4. Sensitivity analysis demonstrated that dry density (*D*) was the most influential parameter with respect to BI.

**Author Contributions:** Conceptualization, M.H. and A.S.M.; methodology, M.J., I.A., M.M.S.S. and M.K.; validation, M.J. and I.A.; investigation, A.S.M. and M.K.; writing—original draft, M.H., A.S.M., M.J., I.A., M.M.S.S. and M.K., writing—review and editing, M.H., A.S.M., M.J., I.A., M.M.S.S. and M.K.; supervision, M.H.; funding acquisition, M.M.S.S. All authors have read and agreed to the published version of the manuscript.

**Funding:** This research was partially funded by the Ministry of Science and Higher Education of the Russian Federation, under the strategic academic leadership program 'Priority 2030' (Agreement 075-15-2021-1333, dated 30 September 2021).

**Institutional Review Board Statement:** Not applicable.

**Informed Consent Statement:** Not applicable.

**Data Availability Statement:** Not applicable.

**Conflicts of Interest:** The authors declare no conflict of interest.

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
