# Peer review of "Predicting Rock Brittleness Using a Robust Evolutionary Programming Paradigm and Regression-Based Feature Selection Model"

_applsci, doi:10.3390/app12147101_

Round 1

Reviewer 1 Report

Please consider the following comments:

  • Check fig. 12, since it is not possible to identify the colors in the legend box.
  • In Fig. 9, it does not make sense to interpolate the points.
  • The description of the Kstar model requires to be improved. Several variables are not defined and the GB parameter is not included in the equations. Provide an example of the instances used in the BI predictions.

Author Response

Response to Reviewer #1

Dear Professor/Dr.

Response: We would like to thank you for reviewing our manuscript. Your comments are all valuable and helpful for revising and improving our paper. We have studied the comments carefully and have made corrections that we hope to meet with your approval. We have studied the comments carefully and have made corrections that we hope to meet with your approval.

Please consider the following comments:

Check fig. 12, since it is not possible to identify the colors in the legend box.

Response: Thank you for your comment. We corrected it accordingly.

In Fig. 9, it does not make sense to interpolate the points.

Response: Thank you for your comment. This kind of plot (trend plot) is a common plot in validation of the ML methods in literature and we drew them one by one, in separately arrangement to better identifying the superior models.

The description of the Kstar model requires to be improved.

Response: Thank you for your comment. We completed this methodology accordingly.

If the examples are real numbers, then it is possible to demonstrate that  is dependent solely on the absolute difference between m and n. Therefore, it can be defined as,

  (10)

where  and  denotes the model parameter and its possible values range from 0 to 1. As a result, the distance between two points is equivalent to their absolute differences. Furthermore, for real numbers, the assumption is that the real space is underlain by a discrete space with extremely short distances between discrete instances. The first thing that has to be done is to evaluate those expressions in their limit as the variable e becomes closer and closer to 0. Because of this, we get:

                                          (11)

The likelihood of generating an integer with a value between i and i+i can be expressed as a probability density function (PDF) as follows:

                (12)

To get the PDF over the real numbers,  can be adjusted in aspects of a real value x.                    (13)

Where , the mean predicted value for x across the distribution P, must be suitable for practical purposes. A number between  and N is picked in the K-Star method, which selects  as the training instance with the lowest distance from m. It should be noted that the KS model was developed in this study by utilizing open-source WEKA software. The main parameter of the KS model is the global blend (GB: n), which is determined by using the trial-and-error method.

Several variables are not defined and the GB parameter is not included in the equations.

Response: Thank you for your comment. We completed some of missing variables.

In LWLR:

"where X is the matrix of input training dataset, W denotes the weight matrix, and Z is the vector of data obtained from the model"

 GB is the abbreviation of global blend (GB). Please refer to previous comment.

Provide an example of the instances used in the BI predictions.

Response: Thank you for your comment. All the testing points are unseen sample and can be considered as an example of BI prediction because none of testing data sets are participate in training process.

Best regards,

Reviewer 2 Report

The authors have tried to develop a prediction of rock using linear genetic programming for estimating the brittleness index. The review article is trying to address the prediction and automation challenges in the geotechnical area. A few comments are mentioned below for the improvement.

- Abstract can be revised by highlighting the novelty part (particularly highlighting the challenge and novelty aspect).

- In general, the prediction models are based on many methods.  In this research manuscript, the comparison with other prediction models are missing. Please include.

- For a research paper, critical analysis and discussion are necessary and the manuscript is lack that. Please expand the discussion.

- Another aspect of the accuracy of the prediction models can be discussed through a different means which is missing. These are very basic to explain the science of variability. 

- The author also highlights the dynamic prediction of these models to practice in real-time operation.

- Each section need a critical analysis of future perspective and research need.

- There are several articles on the subject. The authors need to revisit the reference list. A few of the prolific papers are included below for your reference and request to cite appropriately.

Author Response

Response letter

Ref.: applsci-1711997

Title: Predicting the rock brittleness using a robust evolutionary programming paradigm and regression based feature selection model

Dear Editor,

Thank you for your letter and for the reviewers’ comments concerning our manuscript entitled “Predicting the rock brittleness using a robust evolutionary programming paradigm and regression based feature selection model” (ID: applsci-1711997). We would like to thank the reviewers for thoroughly reviewing our manuscript and for providing many thoughtful comments. Their comments are all valuable and helpful for revising and improving our paper. We have studied the comments carefully and have made corrections that we hope meet with their approval. To make the changes that we made in the manuscript easily identifiable, the changes made to the paper can be identified by the text in red font in the uploaded MS file.

Overview of changes

- Response to Reviewer #1

- Response to Reviewer #2

Best regards,

Response to Reviewer #2

Dear Professor/Dr.

Response: We would like to thank you for reviewing our manuscript. Your comments are all valuable and helpful for revising and improving our paper. We have studied the comments carefully and have made corrections that we hope to meet with your approval. We have studied the comments carefully and have made corrections that we hope to meet with your approval.

The authors have tried to develop a prediction of rock using linear genetic programming for estimating the brittleness index. The review article is trying to address the prediction and automation challenges in the geotechnical area. A few comments are mentioned below for the improvement.

- Abstract can be revised by highlighting the novelty part (particularly highlighting the challenge and novelty aspect).

Response: Thank you for your comment. The abstract has been updated.

- In general, the prediction models are based on many methods.  In this research manuscript, the comparison with other prediction models are missing. Please include.

Response: Thank you for your comment.

In the literature, some studies have been predicted the BI by using different machine learning methods. Yagiz et al. [1] used the genetic algorithm (GA) and particle swarm optimization (PSO) to predict BI. According to their results, the values of R2 ranged between 0.851–0.932. In another study, Koopialipoor et al. [2] predicted the BI through a combination of ANN and firefly Algorithm, and yielded the predictions results with the R2 of 0.896. In the present study, the BI have predicted with a better performance (R2 of 0.953) obtained from the LGP model. This indicates the effectiveness of the model proposed in this study compared to aforementioned models used in the literature.

The above descriptions are added to the results and discussion section.  

1 Yagiz, S., Ghasemi, E. & Adoko, A.C. Prediction of Rock Brittleness Using Genetic Algorithm and Particle Swarm Optimization Techniques. Geotech Geol Eng 36, 3767–3777 (2018). https://doi.org/10.1007/s10706-018-0570-3

2 Koopialipoor M, Noorbakhsh A, Noroozi Ghaleini E et al (2019) A new approach for estimation of rock brittleness based on non-destructive tests. Nondestruct Test Eval 2019:1–22. https://doi.org/10.1080/10589759.2019.1623214

- For a research paper, critical analysis and discussion are necessary and the manuscript is lack that. Please expand the discussion.

Response: Thank you for your comment. We merged both results and discussion sections as "results and discussion". We presented numerous types of analysis such a statistical metric, scatter plot, trend plot, diagnostic analysis in two form of relative deviation and  cumulative frequency percentage which all of them are commonly use in the recent related literature. For satisfying the reviewer, we added some explanation in discussion section.

- Another aspect of the accuracy of the prediction models can be discussed through a different means which is missing. These are very basic to explain the science of variability.

Response: Thank you for your comment. Our main aim was assessing the potential of a robust evolutionary machine learning, namely, linear genetic programming (LGP) for predicting the rock brittleness. About the science of variability, the models used in this study are constructed based on different variables. Before the modeling, we performed a sensitivity analysis and removed several variables from the inputs, and all used variables were the effective parameters on the intensity of BI. According the above explanations, we evaluated the science of variability in our study and after modeling, checked the performance of the models based on statistical functions.       

- The author also highlights the dynamic prediction of these models to practice in real-time operation.

Response: Thank you for your comment. Please consider our explanations.

In this study, 85 data points were collected for modeling the BI which 75% (64 data points) of the data was allocated for the training aim and the rest for the testing aim. The data used in the training phase were completely different with the data used in the testing phase, therefore, the subject of the dynamic prediction can be highlighted. In addition, according to the literature, the graphical validation and comparison tools presented are very good tools to select and evaluate the superior models, and we have applied them in our study.    

- Each section need a critical analysis of future perspective and research need.

Response: Thank you for your comment. We added some future perspective.

According to the objectives of this study, the uncertainty of the data has not been investigated. Uncertainty of data and results of machine learning-based methods due to its great importance can be considered as the subject of future research. Also, the models presented in the current study generally suffer from a lack of laboratory data. Therefore, in the future, it is necessary to examine the accuracy of presented methods with more datasets number.

It should be mentioned that some feature selection methods such as Boruta-random forest can be utilized for specifying the influential parameters which has high ability to capture the non-linear interaction between the predictors and target. This aim can be considered as an alternative of classical sensitivity analysis.

- There are several articles on the subject. The authors need to revisit the reference list. A few of the prolific papers are included below for your reference and request to cite appropriately.

Response: Thank you for your comment. But, any papers have not mentioned.

Best regards,
